# Charge redistribution dynamics in chalcogenide-stabilized cuprous electrocatalysts unleash ampere-scale partial current toward formate production

Feng-Ze Tian [1], Wen-Jui Chang[1], Pei-Jung Liang[1], Yi-An Lai [1], Chia-Shuo Hsu[2], Sheng-Chih Lin[1], Yu-Hsin Chen[3], You-Chiuan Chu[1], Shih-Wen Huang[4] ✉, Hui-Lung Chen [5] ✉ & Hao Ming Chen [1,2,3,6] ✉

Electrochemical $CO_2$ reduction to formate offers a sustainable route, but achieving high selectivity on transition metal catalysts remains a significant challenge, which is typically favored on $p$-block metals. Here, we demonstrate that chalcogenide-stabilized cuprous enables near-complete formate selectivity through a charge redistribution mechanism induced by chalcogenides. Using in situ X-ray absorption spectroscopy, high-energy-resolution fluorescence-detected XAS, Raman, and infrared spectroscopy, we reveal that Cu·chalcogen interactions stabilize $Cu^+$, preventing over-reduction to $Cu^0$ and thereby modulating $CO_2$ adsorption and intermediate binding. This stabilization enhances the *OCHO pathway, shifting product distribution entirely toward formate. CuS exhibits the highest selectivity, achieving a notable 90% faradaic efficiency at −0.6 V and an ampere-scale formate partial current of 1.36 A, demonstrating industrial feasibility. In contrast, CuO, lacking a charge redistribution effect, promotes a mixture of CO and C2 products, underscoring the critical role of chalcogenides in steering product selectivity. This work provides fundamental insights into charge redistribution in $CO_2RR$ and introduces a catalyst design strategy leveraging chalcogen-induced electronic modifications for scalable formate production.

Accompanying the global urgency to achieve emission targets through decarbonization[1,2], electrocatalytic reduction of $CO_2$ ($eCO_2RR$) to fuels and chemicals has gained significant interest in $CO_2$ utilization as a promising avenue for clean fuel production and a valuable resource for industrial applications[3–5]. Formate has been considered an attractive $CO_2RR$ product featuring low energy input with a two-electron pathway and high product market value[6,7]. Electrochemical conversion to formate yields an economically viable pathway with negative net $CO_2$ emissions, while pathways targeting ethanol and ethylene production are unlikely to be sustainable both in terms of profitability and achieving net $CO_2$ emission reductions, even at 100% faradic efficiency[8]. Consequently, the development of formate-selective electrocatalysts has become an intriguing subject of research. Over the past decades, certain formate-selective metal catalysts, such as some

[1]Department of Chemistry, National Taiwan University, Taipei, Taiwan. [2]National Synchrotron Radiation Research Center, Hsinchu, Taiwan. [3]Graduate School of Advanced Technology, National Taiwan University, Taipei, Taiwan. [4]PSI Center for Photon Science, Paul Scherrer Institute, Villigen, Switzerland. [5]Department of Chemical and Materials Engineering, Chinese Cultural University, Taipei, Taiwan. [6]Center for Emerging Materials and Advanced Devices, National Taiwan University, Taipei, Taiwan. ✉e-mail: shih.huang@psi.ch; chl3@ulive.pccu.edu.tw; haomingchen@ntu.edu.tw

p-block metals (Sn, Bi, Pb, and In), have been demonstrated to have high selectivity toward formate production due to their ability to stabilize the key intermediate of *OCHO[9–17]. On the contrary, copper-based catalysts are regarded as non-selective metal catalysts due to the moderate binding energy between copper and $CO_2RR$ intermediates[18]. Most of the research based on Cu catalysts has primarily produced various multi-carbon products like ethylene, ethanol, and n-propanol, while achieving a high selectivity for C1 products, such as formate, proving challenging[6,19]. This challenge stems from the dynamic redox process occurring under cathodic potential, where Cu sites undergo inevitable oxidation and reduction, forming $Cu^0$ and $Cu^{1+}$ sites on the catalytic surface. Such coexistence of $Cu^0$ and $Cu^{1+}$ sites notably promotes the thermodynamics of *CO dimerization, thereby improving the selectivity for multi-carbon products[20–24]. Therefore, the selectivity for electrochemical reduction to formate on a pristine Cu electrode is typically very poor. However, highly selective formate could be achievable upon a copper catalyst with appropriate surface modifications, especially modified by chalcogens. Specifically, sulfurization of copper catalyst drastically enhances the product selectivity toward formate exclusively[25–32]. This crucial discovery has spurred dedicated efforts into mechanistic studies aimed at elucidating how formate emerges as the predominant product over these Cu-derived catalysts.

As of now, the precise catalytic structure of chalcogen-modified copper, especially regarding their mechanism and dynamics for selective reduction, is still unknown and heavily debated. Electronegativity (EN) emerged as a plausible explanation for this observed phenomenon[25], because of the higher affinity between the more electropositive copper and electronegative oxygen within the $CO_2$ molecule, as compared to carbon, the promotion of formate intermediate (*OCHO) is thus preferential. From the standpoint of the greater EN in anion correspondence to increased formate selectivity, reasonably, one might assume that oxide-derived copper catalysts would inherit similar or even greater formate selectivity from copper chalcogenides due to the chemical similarities between oxide and chalcogenides and relatively higher EN in oxygen. Nevertheless, oxide-derived copper tends to produce CO and CO-related products such as $C_2H_4$, ethanol, among others[20,26]. In addition, Strasser et al. proposed that the introduction of halides markedly tuning the selectivity of Cu towards CO and methane is attributed to the increase in negative charge of copper[27], which also contradicts the assumption regarding the EN of anion to selectivity of formate. Furthermore, López et al. asserted that chalcogen adatoms selectively produce formate by either transferring a hydride or tethering $CO_2$[28]. Yet, in most cases, the surface bonding effect is not applicable due to an insufficient sulfur-to-copper ratio to satisfy the required proximity of chalcogen atoms to Cu atoms. Still others suggest high formate selectivity is achieved through the assistance of chalcogen by altering binding energy with intermediates[29–31]. For instance, Shao et al. posited that S−S bonds can moderate the binding energies with various intermediates[31], but it remains an unreasonable assumption because other chalcogen modifications without dichalcogenide bonds could also exhibit a high formate selectivity. In spite of the abovementioned explanations combined with theoretical calculations matching the experimental results, the genuine effect determining selectivity remains a subject of debate. It should be noted that those studies made attempts to clarify the reasons for formate selectivity over copper chalcogenides but lacked specific and direct evidence to better experimentally explicit the selective mechanism. Further endeavors are necessary to better construe the rationale behind the enhanced formate selectivity.

Herein, a series of copper chalcogenides (CuX, X = S, Se, Te) and copper oxide (CuO) were employed as model catalysts to validate the underlying mechanism behind high formate selectivity, in which complete selectivity and an ampere-scale partial current for formate production were achieved through the incorporation of chalcogen anions. Multiple in situ techniques, including X-ray absorption (XAS), surface-enhanced infrared absorption spectroscopy (SEIRAS), and Raman spectroscopies, were integrated with high-energy-resolution fluorescence-detected X-ray absorption spectroscopy (HERFD-XAS) to decipher this elusive preference for formate selectivity. These analyses disclose that chalcogenides induce not only a structural and chemical-state stabilizing effect but also the charge redistribution effect during $CO_2RR$, which could further lead to a dominant formate selectivity over other $CO_2$ products. Our comprehensive investigation may provide valuable insights into the role of copper chalcogenide electrocatalysts in selective $CO_2RR$ towards formate, laying the foundation for the design of selective $CO_2$ reduction catalysts.

## Results
### Structural characterization and electrochemical properties of $CO_2RR$

A series of CuXs was prepared from various chalcogen precursors via hydrothermal synthesis with subsequent thermal treatment. CuO was synthesized in addition, serving as a comparable control featuring group 16 elements (more synthetic details in "Methods"). X-ray diffraction (XRD) patterns of CuS, CuSe, CuTe, and CuO shows diffraction peaks corresponding to crystal phases of Covellite (CuS, PDF#06-0464), Klockmannite (CuSe, PDF#27-0185), Rickardite ($Cu_7Te_5$, PDF#26-1117), and Tenorite (CuO, PDF#44-0706), respectively (Supplementary Fig. 1). The XRD data confirms the presence of the desired crystal phases in the synthesized catalysts (more information in Supplementary Fig. 2, Supplementary Table 1, and Supplementary Note 1). To further evaluate the chemical states and local structure of Cu for as-prepared samples, XAS was utilized, in which the X-ray absorption near-edge structure (XANES) and extended X-ray absorption fine structure (EXAFS) analysis were performed on the Cu K-edge. Supplementary Fig. 3 displays the XANES of CuXs and CuO, along with Cu foil, $Cu_2O$, and CuO as corresponding references. The spectrum of as-prepared CuO matched that of CuO in reference, indicating that the oxidation state is identical to $Cu^{2+}$. The absorption edge of CuXs near 8982 eV lies in the middle of $Cu_2O$ and CuO, suggesting that the oxidation states of Cu in CuXs are below +2, and the relative oxidation state of each sample is sorted out in Supplementary Table 2. Furthermore, as depicted in Supplementary Fig. 4, the $k^2$-weighted Fourier-transformed EXAFS (FT-EXAFS) for as-prepared catalysts displayed radial distances for Cu−S (1.75 Å), Cu−Se (2.05 Å), Cu−Te (2.39 Å), and Cu−O (1.53 Å) in the first coordination shell. These data clearly imply that each Cu center is surrounded by chalcogenide or oxide anions in its local structures, conforming to the XRD observation. Based on the overall structural characterizations above, we can conclude that all of the desired catalysts have a well-defined structural environment with group 16 anions coordinated with Cu cations.

Due to the intrinsic constraint of $CO_2$ mass transport in the H-cell, achieving the scale-up production necessary for industrial applications remains a significant challenge. To address this, the electrochemical performance of $CO_2RR$ was evaluated in 1.0 M KOH solution as electrolyte using a custom-designed flow cell, compatible with a standard three-electrode system and featuring a continuous supply of $CO_2$ (Supplementary Fig. 5, more details in "Methods"). Linear sweep voltammograms (LSV) of CuXs, as well as CuO are shown in Supplementary Fig. 6. Gas chromatographic (GC) analysis for gas-phase products and nuclear magnetic resonance (NMR) analysis for liquid products were employed under electrolysis applied with the chronoamperometry method at different cathodic potentials (Supplementary Fig. 7). The faradaic efficiency (FE) profiles of the electrochemical $CO_2RR$ for various catalysts are demonstrated in Supplementary Fig. 8. It was found that all of the CuXs produced formate ($HCOO^-$) as their only dominant $CO_2RR$ products, as evidenced by the product profiles. By sharp contrast, CuO exhibited a wide $CO_2RR$ product distribution, including a reckoned amount of carbon monoxide, hydrocarbons, and alcohols. From the point of view of $CO_2RR$ selectivity, it is obvious that

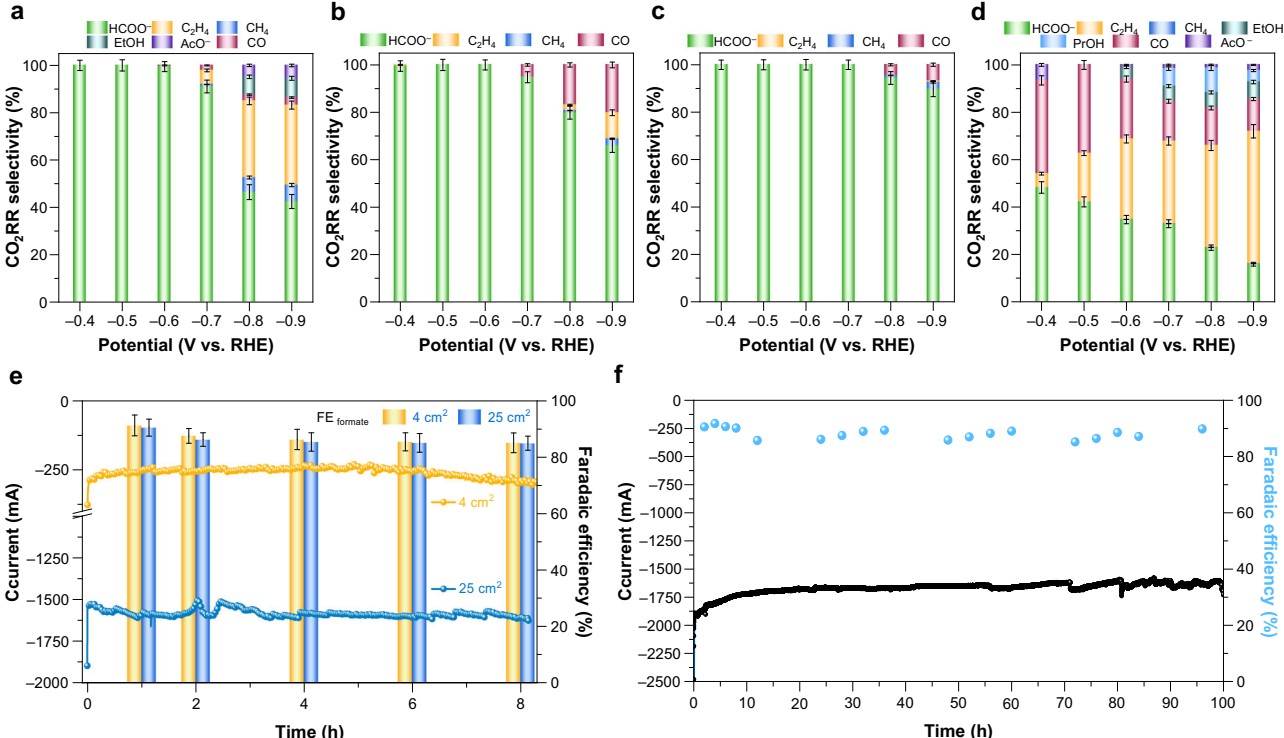

**Fig. 1 | Electrocatalytic performance of CO₂RR. a–d** CO₂RR selectivity as a function of potential for (**a**) CuS, (**b**) CuSe, (**c**) CuTe, and (**d**) CuO. CO₂RR selectivity was calculated as the percentage of each product, excluding the contribution of H₂. **e** Productive performance toward formate for CuS. **f** Electrochemical CO₂RR stability of CuS recorded at −0.6 V$_{RHE}$. All potentials are reported vs RHE with 85% *iR*-correction. The error bars represent the standard deviation of three independent measurements. Source data for Fig. 1a–f are provided in the Source Data file.

the selectivity of formate is clearly influenced by the coordination environment of the metal center (Fig. 1a–d). All of CuXs delivered a nearly complete CO₂RR selectivity towards formate under optimum potential window (−0.4 V to 0.7 V vs RHE) (Fig. 1a–d and Supplementary Fig. 37a). Yet, subsequently increasing the applied potential from −0.7 V to a higher cathodic potential, FE and selectivity of formate started to descend. On the other hand, as displayed in Fig. 1d, CuO exhibited no CO₂RR selectivity, yielding a variety of carbon-based products, mainly ethylene, formate, and CO with a trace amount of alcohols. In bright contrast to other CuXs, CuO showed no specific product selectivity, which confirms the suspicion made earlier in the introduction section. As evidenced by the product profile and CO₂RR selectivity, we note that such exclusive formate production seems to occur within a specific coordinated environment only—specifically, the chalcogenide in conjunction with Cu (Cu−S, Cu−Se, and Cu−Te).

Thereafter, to further assess the productive performance of formate generation, the electrode area was scaled up to demonstrate the capability of mass formate production using chalcogenide-based copper. Across certain applied potential range from −0.4 V to −0.7 V, the FE of formate had a tendency in an order of CuS (highest, ~90%) ≫ CuSe (~36%) > CuO (~27 c%) > CuTe (lowest, ~22%) (Supplementary Fig. 8). The partial current density of formate (*j*(HCOO⁻)) corresponding to the highest FE for formate also displayed a volcano trend with CuS being the most active for formate production (~59.6 mA cm⁻² at −0.6 V), significantly outperforming all others under the same condition (Supplementary Fig. 9). The chronoamperometric results of CuS at −0.6 V (Fig. 1e) demonstrated that the partial current of formate exceeded the ampere-scale (1.36 A), with formation rate estimated at approximately 25.4 mmol h⁻¹. To evaluate the operational durability, we adopted a gas diffusion layer pretreatment strategy aimed at enhancing electrode hydrophobicity and mitigating electrowetting[32]. This approach enabled us to conduct a prolonged CO₂RR stability test for 96 h, during which the FE for formate

consistently remained above 85% (Fig. 1f). These results reveal that, in our system, chalcogenide-derived copper is achievable for high formate production with relatively high selectivity, highlighting the effectiveness of manipulating product selectivity through coordination modification of copper, which showcases compatible feasibility for industrial-scale implementation.

## In situ identification of CO₂RR intermediates
To visualize the above-discussed selective and unselective formate generation process on the CuXs and CuO samples, in situ Raman and SEIRAS were performed to track the signals of key intermediates, namely, *CO and *OCHO[33]. The in situ Raman and SEIRAS spectra of CuS, representative samples of CuXs, and CuO are displayed in Fig. 2 (more details in "Methods" and Supplementary Note 2). As shown in Fig. 2a, formate-related peaks at Raman shift of 1300–1500 cm⁻¹ were constantly observed for the formate-selective CuS with cathodic potential applied under the CO₂RR electrolysis[34]. A doublet peak at 1440 cm⁻¹ and 1460 cm⁻¹ is verified as the identity of the oxo-bridged site of *OCHO[35,36], while a broad band centered at 2800 - 3000 cm⁻¹ signifies the characteristic peak of C−H stretching modes, likely associated with adsorbed formate on copper[37]. Notably, no additional distinct Raman peak, particularly C≡O stretching mode near 2000 cm⁻¹, was observed throughout in situ measurements, suggesting a highly selective pathway favoring *OCHO formation over *CO intermediates (Supplementary Fig. 12). As demonstrated in Fig. 2b, the intensities of formate-related peaks became increasingly pronounced with the rise in applied cathodic potential, suggesting a concurrent increase in the *OCHO population. Notably, the peak intensity of 1440-1460 cm⁻¹ reached a maximum situation at the potential of −0.5 V to −0.6 V, coherent with the potential of highest FE for formate production on CuS. Subsequently, beyond the inflection potential of −0.6 V, a drastic decline in peak intensity was observed, implying the reduction in *OCHO population in tandem with the changes in product

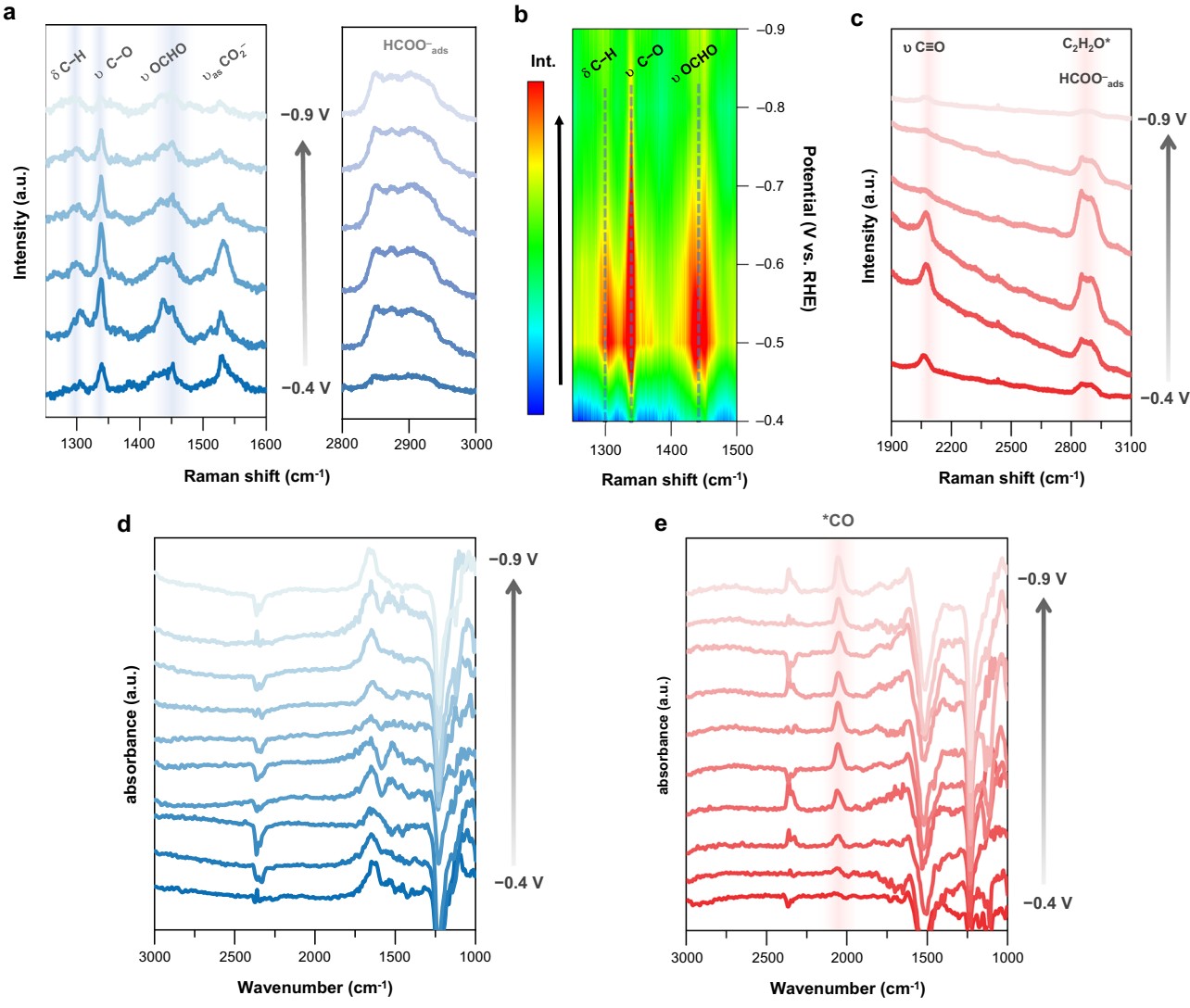

**Fig. 2 | In situ Raman spectroscopy and SEIRAS for studies of CO₂RR intermediates. a–c** In situ Raman spectra as a function of potential for **a**, **b** CuS and **c** CuO. **d**, **e** In situ SEIRAS spectra as a function of potential for **d** CuS and **e** CuO. All potentials are reported vs RHE with 85% *iR*-correction. Source data for Fig. **a–e** are provided in the Source Data file.

profile. Regarding CuO, a distinct Raman peak corresponding to the C≡O stretching mode was observed at 2075 cm⁻¹ under identical conditions as those for CuS (Fig. 2c)[37]. In addition, a broad C–H stretching band around 2800–3000 cm⁻¹ was detected, likely arising from adsorbed intermediates such as formate (HCOO⁻ₐds) and possible C2 intermediates ($C_2H_2O^*$) formed via C–C coupling and hydrogenation[38]. The intensified C–H stretching signals with negative potentials imply an accumulation of C–C coupling intermediates, consistent with the observed product profile (Fig. 1d). At potentials beyond −0.7 V, a decreased intensity for the C–H stretching band was noted, likely due to the desorption or conversion of these intermediates under stronger reductive conditions. Such copresence of *CO, *OCHO, and $C_2H_2O^*$ intermediates represents the unfavorable selective reduction route for CuO, while similar results were also observed on SEIRAS spectra (Figs. 2d, e for the CuS and CuO, respectively). As depicted in Fig. 2d for CuS, the formate-related peak could not be distinctly observed due to the surface selection rule affecting C–H stretching signals at 2800–3000 cm⁻¹, as well as peak overlap among the H–O–H bending mode of adsorbed water, C–O stretching of $HCO_2^-$, and COO⁻ symmetric stretching at around 1300–1600 cm⁻¹[39,40]. Nevertheless, the signals related to CO remained silent in the range of 1800–2100 cm⁻¹, aligning with the results obtained from Raman spectra. In comparison,

for CuO, Fig. 2e reveals an obvious absorption peak at 2051 cm⁻¹ attributed to CO stretching in atop-adsorbed CO, in line with the Raman result as well[41]. Consequently, the above electrochemical and Raman spectral characterization concludes that the incorporation of chalcogenides over Cu can significantly facilitate the highly selective CO₂RR toward formate.

## Dynamic chemical state and structural evolution under CO₂RR

Despite the successful identification of reaction intermediates being evident for potential differences in selective mechanism between CuXs and CuO, the interplay between chemical states of Cu and structural variations serves as a more relevant indicator for catalytic behavior and CO₂RR selectivity[42–45]. Further characterizations were enforced to explore the connection between catalytic structure and formate selectivity during CO₂RR. As the chemical states of copper have proved their influences on the CO₂RR selectivity[22,46], in situ XAS experiments at the Cu K-edge were conducted to track the Cu oxidation states under operational conditions. In Supplementary Fig. 13, in situ Cu K-edge XANES spectra revealed the electronic structure of each catalyst under varying cathodic potentials, while the relative Cu oxidation states could be derived from the first derivative of normalized absorbance as a function of energy, as illustrated in Fig. 3a–d. Upon applying cathodic

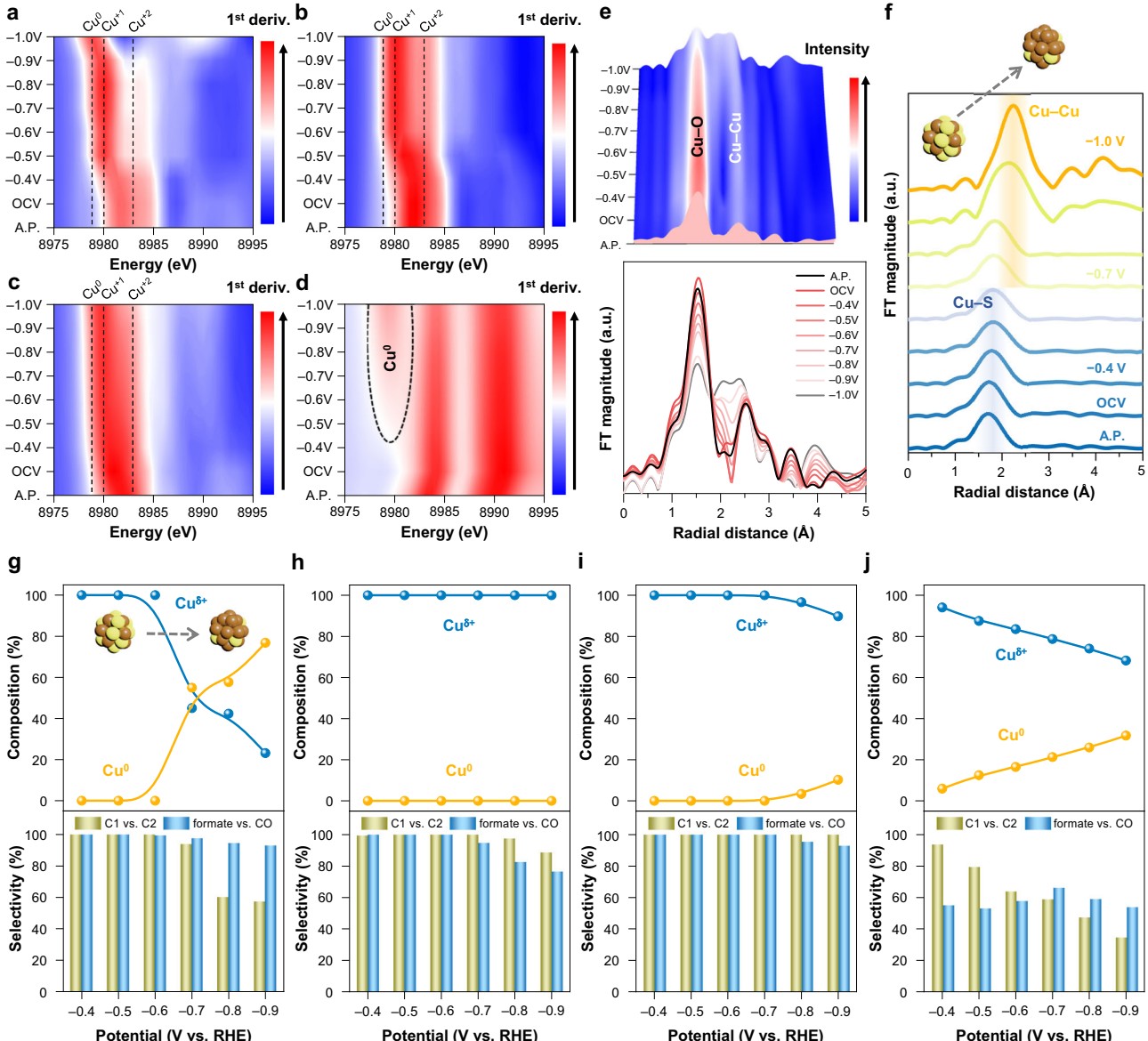

**Fig. 3 | In situ Cu chemical state and dynamic structure analysis. a–d** The first derivative of normalized absorbance in in situ Cu K-edge XANES as a function of potential for **a** CuS, **b** CuSe, **c** CuTe, and **d** CuO. AP as-prepared electrocatalysts, OCV open circuit potential. **e, f** In situ $k^2$-weighted FT-EXAFS of Cu K-edge for **e** CuO and **f** CuS. **g–j** Evolution of Cu composition (top lines) and product selectivity (bottom bars) with potential for **g** CuS, **h** CuSe, **i** CuTe, and **j** CuO. All potentials are reported vs RHE with 85% *iR*-correction. Source data for Fig. 3a–j are provided in the Source Data file.

potentials from open circuit potential (OCV), CuXs underwent a slight reduction and eventually stabilized to locate their maximum peak intensities at ~8980 eV, corresponding to an oxidation state of +1. Conversely, as evidenced by the oxidation state analysis deconvoluted through linear combination fitting using empirical standards of Cu foil, $Cu_2O$, and CuO (Supplementary Figs. 14–22 and Supplementary Tables 4 and 5), CuO exhibited a continuous and dynamic reduction process, transitioning from $Cu^{2+}$ to $Cu^0$ and possessing a consistently higher average chemical state as compared to CuXs while the overall variation in oxidation state for CuXs and CuO under different operational conditions is summarized in Supplementary Fig. 23.

Afterward, in situ FT-EXAFS of Cu K-edge was analyzed for probing the dynamic catalytic structure. The coordination environment of CuO and CuS under various operational conditions are presented in $k^2$-weighted FT-EXAFS (Fig. 3e, f, respectively). In Fig. 3e, the structure of CuO underwent a continuous and dynamic variation throughout the entire $CO_2RR$ electrolysis. The Cu–O bond at about 1.53 Å experienced

a successive decrease in peak intensity, while radial distance in approximately 2.0 Å, presumed to be the Cu–Cu bond in the metallic Cu cluster, exhibited a totally inverse behavior as Cu–Cu peak intensity rose simultaneously with the elevated cathodic potential. This observation in R-space of CuO identifies the reduction of Cu–O into a metallic Cu–Cu bond, suggesting the composition of as-prepared CuO gradually shifts towards a combination of CuO with a portion of a metallic Cu cluster. The FT-EXAFS data align with the chemical state analysis obtained from in situ XANES that a portion of $Cu^0$ species was detected from linear combination fitting. Regarding CuS, Fig. 3f presents the stacked $k^2$-weighted in situ FT-EXAFS. Notably, different from the R-space spectra of CuO, the Cu–S bond at approximately 1.75 Å preserved its original feature without the formation of new bonding in comparison to Cu–O and metallic Cu–Cu bond in CuO in the range from OCV to −0.6 V, confirming no other Cu cluster species was generated under a potential of below −0.6 V (more fitting details in Supplementary Figs. 24–27 and Supplementary Table 6). Comparable

phenomena were also observed in FT-EXAFS of CuSe and CuTe (Supplementary Fig. 28), where Cu−Se and Cu−Te bonds were ultimately retained with no production of metallic Cu−Cu bond. These results imply that CuXs are stabilized to withstand the reductive driving force within a suitable potential window, preventing $Cu^{1+}$ from further reduction to metallic $Cu^0$. However, in the case of CuS, once the cathodic potential greater than −0.7 V was applied, a similar denouement of CuO was found. A pronounced peak around 2.0 Å (Supplementary Fig. 26), ascribed to the metallic Cu−Cu bond, began to intensify with a higher applied potential. Supplementary Fig. 27 also displays the transformation of Cu−S into Cu−Cu bond at the region of cathodic potential above −0.7 V, indicating the presence of a metallic Cu cluster. To corroborate these structural changes, in situ XRD was deployed (Supplementary Fig. 29). As displayed in Supplementary Figs. 30−31, unexpectedly, no structural transformation or formation was observed, including in CuO. We suspect that this apparent absence of structural change may stem from the limited duration of the in situ XRD measurements, which could have been insufficient to capture slower phase transitions.

As the preceding results imply a potential influence of temporal effects on structural evolution, it is essential to distinguish between time-dependent and potential-dependent behaviors. Short-term in situ studies may not adequately reflect the long-term dynamics pertinent to practical electrolysis[47]. To directly probe the structural evolution during extended operation, we further performed real-time in situ XAS on the 96-h electrochemical stability test. As shown in Supplementary Fig. 32a−c, the XANES spectra of CuXs at −0.6 V displayed slight spectral changes associated with initial electrochemical activation but remained stable thereafter, consistently indicating the preservation of the $Cu^+$ oxidation state. CuO, on the other hand, underwent rapid and substantial reduction, showing significant spectral changes as early as 12 h and complete transformation to metallic Cu by 24 h (Supplementary Fig. 32d). EXAFS analysis further revealed negligible structural changes for CuXs throughout electrolysis at −0.6 V (Supplementary Fig. 33a−c), whereas CuO displayed the emergence of Cu−Cu bonding features, indicating the formation of metallic Cu domains (Supplementary Fig. 33d). To validate these findings, ex situ XRD was also performed at various electrolysis intervals under −0.6 V (Supplementary Fig. 34). Contrary to initial in situ XRD results (Supplementary Figs. 30−31), ex situ XRD analysis revealed that CuO exhibited significant structural transformation into metallic Cu phase. Conversely, CuXs largely retained their original crystallographic structures throughout 96-h operation, consistent with the in situ XAS data. This discrepancy supports the notion that the previously undetected changes in in situ XRD may be attributed to the insufficient measurement duration. Additional characterizations, including high-resolution transmission electron microscopy (HR-TEM) and inductively coupled plasma mass spectrometry (ICP-MS), were performed on both pre- and post-electrolysis samples. HR-TEM images for CuXs in Supplementary Fig. 35a−c show well-preserved lattice fringes after 96 h at −0.6 V, while CuO demonstrates pronounced structural transformation with the formation of metallic Cu domain (Supplementary Fig. 35d). Additionally, the atomic retention of chalcogenide in CuX electrodes was estimated by quantifying the concentration of dissolved anions in the electrolyte through ICP-MS analysis (Supplementary Fig. 36). All CuXs exhibited robust compositional stability over prolonged electrolysis with minimal chalcogenide leaching for CuSe and CuTe (<3%) and slight to moderate leaching for CuS (~20%). This observed extent of sulfur loss is presumably attributed to the partial dissolution of one sulfur atom from the disulfide ($S_2^{2-}$) moiety. Nonetheless, this degree of leaching did not result in any observable structural reconstruction, as corroborated by in situ XAS, ex situ XRD, and HR-TEM analysis. Evidentially, although the cleavage of some disulfide bonds results in the loss of a small number of sulfur atoms, the terminal S−Cu bonds are retained, thereby preserving the local coordination environment around Cu. Collectively, these observations confirm that the structural integrity of CuXs can be reliably preserved over extended operation, primarily governed by potential-dependent rather than time-dependent control, whereas CuO inherently manifests inferior stability and is prone to irreversible structural reconstruction under reductive conditions.

## $CO_2RR$ pathway and $Cu^{\delta+}$ stabilization

Certainly, discerning the linkage between dynamic catalytic structure and product selectivity is essential for unfolding the mechanism of selective reduction toward formate. Accordingly, the evolution of Cu composition and product selectivity as a function of potential is presented in Fig. 3g−j. The composition of $Cu^0$ was estimated by coordination environment in FT-EXAFS and linear combination fitting, while the percentages of oxidized $Cu^{\delta+}$, including $Cu^+$ and $Cu^{2+}$, was evaluated from the average Cu oxidation state in Supplementary Fig. 23. Besides, by further categorizing the overall product selectivity into the selectivity of each representative product as a function of potential, the catalytic behavior of each catalyst can be readily dissected. The representative products chosen for this analysis include mono-carbon (C1) products, specifically formate and CO, as well as multi-carbon (C2) products, primarily ethylene (Supplementary Figs. 37−39). In the initial analysis, our focus centered on the selectivity toward C1 and C2 products, as delineated in Fig. 3g−j. In the potential range from −0.4 V to −0.7 V, C1 products predominate over C2 products across all CuXs, nearly reaching 100%. On the contrary, CuO exclusively shifts C1 selectivity toward C2 products in a manner that exhibits an almost linear correlation with the potential; as C1 selectivity drops, C2 selectivity rises complementarily. Consequently, the reasons behind the high C1 selectivity for CuXs and the inversion between C1 and C2 selectivity for CuO were evidently uncovered from the relative composition between $Cu^{\delta+}$ and $Cu^0$. The high C1 selectivity for CuXs primarily originated from the anticipated prevalence of non-$Cu^0$ mixed-valence species, with chalcogenide-stabilized $Cu^{\delta+}$ being the dominant species directing selectivity toward C1 products. Augmenting this bolstering in the $Cu^{\delta+}$ stability is presumably attributed to the intrinsically elevated energy levels of chalcogenide $p$ orbitals with increasing principal quantum numbers ($3p$, $4p$, and $5p$), which raise the energy of antibonding states in CuXs and thereby strengthen the Cu−X bonds by reducing antibonding orbital occupancy[48]. To further support this claim, density functional theory (DFT) calculations on the projected density of states (PDOS) were introduced (Supplementary Fig. 40). One can evidently find that CuO features antibonding states located closer to the Fermi level, facilitating their population under $CO_2RR$ potentials and rendering the Cu−O bond more susceptible to cleavage. In contrast, CuXs display higher-energy antibonding states due to chalcogenide ligands with higher principal quantum numbers, leading to reduced occupancy of antibonding orbitals and greater Cu−X bond stability under $CO_2RR$ conditions. Although CuO seems to manifest stronger orbital interactions, as evidenced by its more pronounced splitting, its antibonding states reside at lower energy levels, which fundamentally compromise the stability of the Cu−O bond. These results suggest that the incorporation of chalcogenides with higher $p$ orbital energy levels can effectively diminish antibonding interactions within the X $p$ and Cu $3d$ hybridization, thereby reinforcing the stability of the $Cu^{\delta+}$ sites. Therefore, the Cu−X bonds, unlike the Cu−O bond, are preserved under certain operational conditions. This circumvents the presence of mixed-valence Cu, enabling CuXs to retain their product selectivity.

In contrast, the presence of $Cu^0$ in CuO was significant and exhibited a linear increase with applied cathodic potential, driven by the dynamic filling of electrons into the antibonding orbitals of CuO. The higher antibonding orbital occupancy induced structural instability in CuO under cathodic conditions. Notably, the reduction in $Cu^{\delta+}$ content, transitioning to that of $Cu^0$, shared a similar trend with

the decline in C1 selectivity. It is likely due to the coexistence of $Cu^{\delta+}$ and $Cu^0$ mixed-valence Cu activating the multi-carbon reaction pathway, facilitating C−C coupling for the production of C2 products, and ultimately resulting in a diverse product distribution with low selectivity[49–51]. Moreover, the C1 selectivity of CuS endured a drastic descent at around −0.7 V, which is in line with the observed transformation in Cu K-edge FT-EXAFS. Once beyond −0.7 V, C1 selectivity in CuS was prone to transition to C2 products, similar to the catalytic behavior of CuO, which could be attributed to the formation of a majority of $Cu^0$ cluster owing to the evacuation of sulfide. As compared to CuO, the relatively weaker antibonding interaction in CuS delayed the occurrence of Cu−S cleavage to higher cathodic potentials. With even weaker antibonding interactions in the context of CuSe and CuTe, this trend is of trifling impact, as substantiated by in situ XAS (Supplementary Fig. 28), fortuitously enabling the preservation of C1 selectivity.

Lastly, we directed our attention to the selectivity between formate and CO within the C1 products. For CuO, there is no clear preference for CO or formate, with the formate-to-CO selectivity ranging between 50% and 65%. In stark contrast, CuXs possess complete dominance in formate selectivity over CO, with selectivity consistently surpassing 80%. Notably, the formate-to-CO selectivity of CuO increases with the applied cathodic potential, reaching a maximum at −0.7 V. This inflection point in CuO coincides with the descending point in CuXs, implying a shared factor contributing to the inversion in selectivity. To disclose this inversion, analyzing the individual Tafel slopes for each catalyst revealed distinct turning points at −0.7 V (Supplementary Fig. 41). The pronounced change in the Tafel plot beyond −0.7 V could be interpreted as a significant potential bias leading to modifications in the binding energies of intermediates, consequently redistributing the product selectivity. Nonetheless, there are still no clear clues explicating the selectivity between formate and CO. The genuine selective mechanism that propels the complete formate selectivity over CO for CuXs remains an unsolved issue. To delve a step deeper, the fundamental factor dictating the preference for formate selectivity over CO is imperatively required to be uncovered.

### Dynamic frontier orbital evolution through in situ HERFD-XAS and mechanistic investigation

All results above inspire us to further investigate the underlying driving force that inclined CuXs toward formate. Since the nearly-filled Cu $3d$ orbitals were predominantly engaged in binding with anions, the vacant Cu $4p$ orbitals are considered the primary orbital responsible for the activation of the $CO_2$ molecule. Due to the more diffuse nature and greater spatial extension of the nucleus as compared to $d$ orbitals, $4p$ orbitals are more prominently involved in interactions with larger, delocalized orbitals, such as the $\pi^*$ orbitals in $CO_2$. Typically, the activation process involves the overlap of Cu $4p_z$ orbital, oriented perpendicular to the surface plane, with $\pi^*$ orbital of $CO_2$ along the z-axis. Hence, the dynamic behavior of Cu $4p$ orbitals (frontier orbitals of Cu) is pivotal in dictating the reaction pathway. With the energies of the $4p_x$, $4p_y$, and $4p_z$ orbitals contingent upon the specific electronic transitions occurring near the absorption edge, the contribution of these $4p$ orbitals defines the XANES features of Cu K-edge[52]. Regardless of previous in situ XAS results have already provided valuable information, including the chemical states and structural transformation, the conventional total fluorescence yield (TFY) spectra still suffer from insufficient energy resolution resulting from the intrinsic broadening of core-hole lifetime[53,54]. The subtle information of the $1s \rightarrow 4p$ transition regarding $4p_x$, $4p_y$, and $4p_z$ orbitals could not be effectively deconvoluted. To surmount this limitation and gain real-time insight into dynamic orbital characteristics, in situ Kα high-energy-resolution fluorescence detected XAS (HERFD-XAS) was utilized to characterize the frontier orbitals on reactive Cu[55,56]. The setup of in situ HERFD-XAS

measurements is presented in Fig. 4a. (more details in "Methods"). The in situ HERFD-XAS for CuXs and CuO under $CO_2RR$ are displayed in Fig. 4b. All HERFD-XAS spectra exhibit distinct two-band features at the main edge region (8979–8987 eV). The relatively lower energy band ranging from 8980 eV to 8982 eV should correspond to the transition of Cu $1s$ electrons to the low-lying $4p_z$ orbital, whereas the higher energy band near 8982–8987 eV represents the transition from $1s$ to $4p_{x,y}$[52]. To support the interpretation of these spectral features, we integrated DFT calculations of PDOS on Cu $4p$ orbitals with the experimental spectra, as shown in Supplementary Fig. 42. The PDOS results are in strong agreement with these assignments, confirming that the first feature is primarily derived from Cu $4p_z$ states, while the second feature arises predominantly from $4p_{x,y}$ orbital contributions.

Once subjected to apply potential at −0.6 V, both $4p_z$ and $4p_{x,y}$ peak intensities in all catalysts, except CuTe, encountered a decline as compared to the initial spectra. This decline could be attributed to the increased electron population during electrochemical conditions, resulting in elevated electron densities of $4p$ orbitals. The spectral change was not obvious for CuTe because of the relatively minor difference in the chemical state between as-prepared and operational conditions. One can evidently find that the area under edge absorption for CuO, both with and without applied potential, was identified to be significantly smaller than CuXs. This observation implies that CuXs constantly exhibit comparably higher degrees of Cu $4p$ orbital vacancy. Intriguingly, the intensities of the $4p_z$ bands diverge distinctly among CuXs and CuO, where CuXs still retain a higher intensity in the $1s$ to $4p_z$ transition despite a slight decrease after applying cathodic potential; meanwhile, CuO distinctly exhibits a significant drop in intensity, with the $4p_z$ band nearly vanished. We explicated that this spectroscopic disparity may arise from different orbital configurations as a result of their respective coordinated environment. Specifically, the alignment of $4p$ orbitals could be deduced from the first derivative of normalized absorbance as a function of energy (Supplementary Fig. 43). Figure 4c specifies the energy position of $4p_z$ and $4p_{x,y}$ bands for CuX and CuO before and during $CO_2RR$, which presents the sketch of the dynamic Cu $4p$ orbital configuration. From as-prepared conditions, energy levels followed the order of CuXs $4p_z$ < CuO $4p_z$ < CuXs $4p_{x,y}$ < CuO $4p_{x,y}$ with a significantly larger orbital splitting of the $4p$ orbitals for CuO, reflecting stronger Cu−O interactions and resulting in increased electron population in the low-lying $4p_z$ orbital. Contrariwise, heavier chalcogens interact more weakly with Cu, leading to smaller orbital splitting energy as observed in CuXs, which allows for a more delocalized and balanced distribution of electron density across both the $4p_z$ and $4p_{x,y}$ orbitals. In agreement with these observations, PDOS calculations further substantiate that CuO possesses a markedly greater energy separation between the $4p_z$ and $4p_{x,y}$ states, whereas CuXs display a progressive decrease in orbital splitting across the series. Consequently, CuXs exhibit greater $4p_z$ orbital vacancies at relatively lower energy, endowing their frontier Cu $4p_z$ orbitals with enhanced electropositivity. We posit that such a reversal in $4p_z$ electropositivity for CuO and CuXs paves the way for adaptive control over $CO_2RR$ intermediates. The attenuated orbital engagement in chalcogenides crucially modulates the electron density in the Cu $4p_z$ orbital to adopt two different binding orientations, leading to the formation of either Cu−OCHO* or Cu−COOH* intermediates, which eventually lead to the production of formate and CO, respectively. Based on this, we further propose a mechanism for formate-selective $CO_2RR$, emphasizing this effect described as charge redistribution on Cu $4p_z$ orbitals. In the case of CuXs, the relatively electropositive Cu tends to interact with the electron-rich O atom of the $CO_2$ molecule to form Cu−OCHO*. In the scenario of CuO, the comparatively less electropositive Cu exhibits no significant bonding preference for the O atom, instead primarily forms a bond with the electron-deficient C atom of the $CO_2$ molecule to generate Cu−COOH*, as depicted in Fig. 4c.

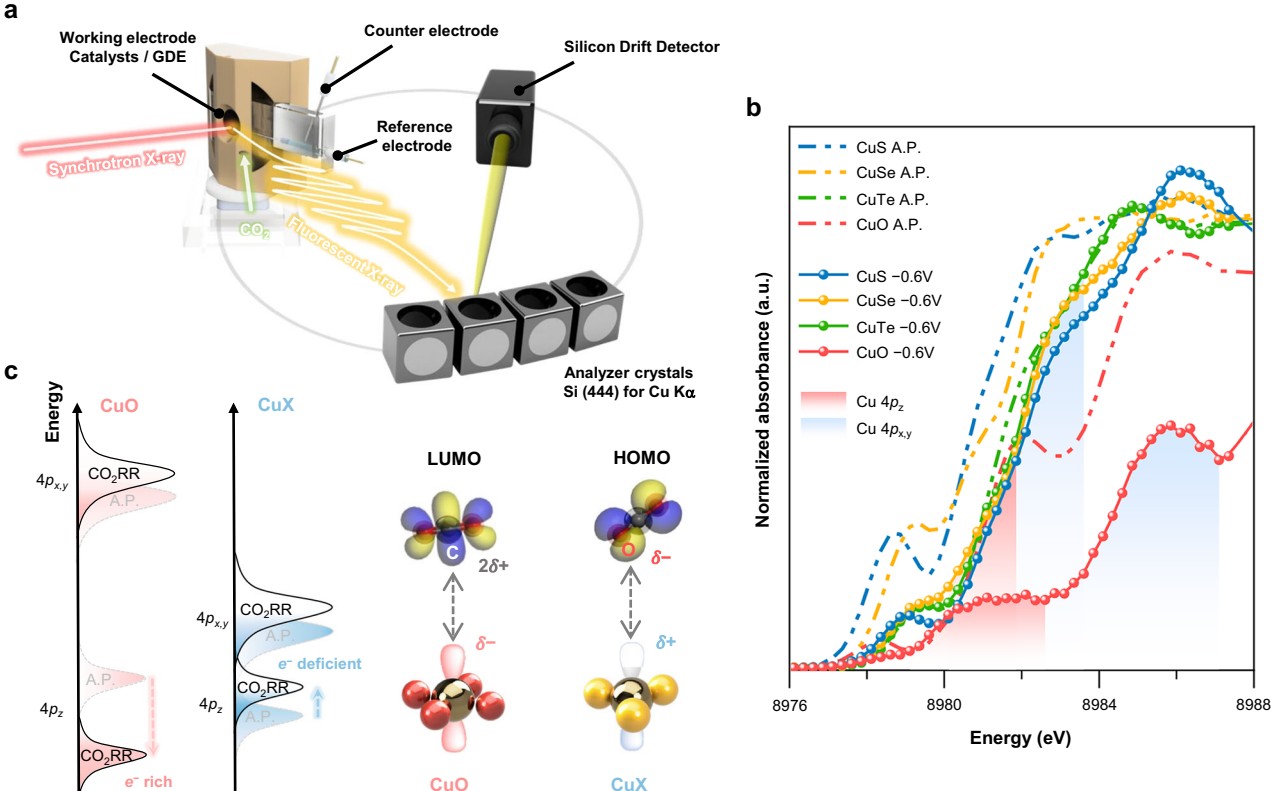

**Fig. 4 | In situ HERFD-XAS and mechanistic investigation of formate selective CO$_2$RR. a** Schematic representation of the in situ HERFD-XAS facility with an electrochemical gas diffusion flow cell. **b** In situ HERFD-XAS of Cu K-edge for CuXs and CuO. All potentials are reported vs RHE with 85% *iR*-correction. **c** Sketch of Cu 4*p* orbital alignment derived from the first derivative of in situ HERFD-XAS (left) and proposed scheme for the charge redistribution induced orbital interactions for corresponding CO$_2$RR intermediates (right). Source data for (**b**) are provided in the Source Data file.

One should notice that the orbital alignment undergoes significant variation under operational conditions, underscoring the distinct orbital interactions during electrocatalytic reduction (Fig. 4c). The 4$p_z$ orbital of CuO experienced an obvious downshift toward lower energy compared to the initial condition, whereas those of CuXs generally either maintain their energy levels or undergo a slight upshift. This behavior could be related to the dynamic engagement of frontier Cu 4$p$ orbitals with CO$_2$ during operational conditions. The observed downshift in the 4$p_z$ energy level for CuO could be attributed to the interaction with the LUMO (π* orbital) of CO$_2$, wherein electron density is partially delocalized from the 4$p_z$ orbital, stabilizing its energy state. As depicted in Fig. 4c, the delocalized nature of the π* orbital of CO$_2$ enables interaction with electron-rich 4$p_z$ orbitals of CuO in an optimal geometric configuration through the C atom of CO$_2$, facilitating effective π orbital-mediated charge delocalization. This dynamic downshift, observed exclusively in CuO, suggests the prevalence of Cu−COOH* promoted by this delocalization process. The product selectivity of CuO further corroborates our claim, as the respective products derived from Cu−COOH*, including multi-carbon products, rise progressively with larger cathodic potential. Conversely, the presence of reverse shifts in CuXs may involve alternative orbital interactions beyond direct overlap with the π* orbital of CO$_2$. The upshift in energy level of the 4$p_z$ orbital could result from the interaction of the HOMO (lone pair on the O atom) of CO$_2$, while the σ donation from the HOMO on the O atom destabilizes and elevates the 4$p_z$ energy state prior to weaker and less direct orbital overlap with the π* orbital of CO$_2$ through the O atom. This interpretation of inverse shifting in CuXs signifies the dominant population of Cu−OCHO*. The magnitude of energy upshifts exhibits a certain correlation with the FE of formate, both of which follow an ascending trend from CuTe to CuS.

Another correlation, irrespective of the direction of orbital energy shifting, can be observed within FE of H$_2$ as well. Particularly, CuS and CuO, which demonstrate greater deviations from their initial orbital energy states, were found to have lower proportions of HER competition compared to CuSe and CuTe. The extent of HER is inversely correlated with the magnitude of 4$p_z$ energy shifting. This phenomenon may be explained by the fact that proton adsorption is more strongly influenced by the *d* band states of Cu instead of the frontier 4$p$ orbitals, where the *d* orbitals on Cu are more localized and denser, being spatially closer to the nucleus and possessing higher electron density near the surface. Such localization makes them more effective at directly interacting with adsorbates, protons (H$^+$), which require high electron density for bonding. The computational insights derived from PDOS calculations (Supplementary Fig. 40) indicate that the Cu 3$d_{z2}$ band gradually shifts toward the Fermi level along the periodic progression from CuS to CuTe, thereby enhancing its overlap with the 1$s$ orbital of the proton. Accordingly, a greater proportion of proton adsorption relative to CO$_2$ activation necessitates less involvement of 4$p$ orbitals in the bonding process. As a result, the 4$p_z$ orbital would encounter progressively weaker forces for driving energy deviations as one moves periodically from CuO to CuTe.

From the perspective of chalcogen anions, we conducted further investigations into chemical environments through ex situ S K-edge and in situ Se and Te K-edge XAS. Because of the low absorption energy at the S K-edge, 2472 eV, implementing in situ methodologies poses challenges in the context of tender-XAS for CO$_2$RR. Accordingly, ex situ S K-edge XAS was selected as the primary alternative. The ex situ S K-edge XANES of CuS is depicted in Supplementary Fig. 44, while Supplementary Fig. 45 presents the $k^2$-weighted in situ FT-EXAFS for CuSe and CuTe, providing insight into the coordination

environment from the viewpoint of chalcogen atoms during $CO_2RR$. Regarding the in situ Se and Te K-edge FT-EXAFS, Se−Cu and Te−Cu bonds at approximately 2.06 Å and 2.17 Å were consistently observed (Supplementary Fig. 45). From the ex situ S K-edge XANES the pre-edge peaks of as-prepared and OCV are identical to the characteristic absorption of Covellite CuS as illustrated in Supplementary Fig. 44. The overlap of orbitals between a 3 *d* valence hole in Cu with a suitable S $3p$ orbital gives rise to a pre-edge feature in the S K-edge spectrum, which serves as an ideal signature for monitoring the electronic structure of both Cu and S[57]. Upon applying a cathodic potential, a significant change in the pre-edge feature occurred from OCV to −0.4 V. Within a mild potential range of −0.4 V to −0.6 V, the pre-edge feature at 2471.3 eV, corresponding to the energy difference between S 1*s* orbital and $S_2^{2-}$ $3p$ orbital delocalized with the $Cu^{\delta+}$ hole, diminished, implying the conversion of disulfide species ($S_2^{2-}$) to $S^{2-}$ as a result of the loss of $S_2^{2-}$-related absorption[58]. Still, a pre-edge peak at 2470.1 eV, indicative of the transition of S 1 *s* to $S^{2-}$ $3p$ orbital delocalized with the $Cu^{\delta+}$ hole, persists, representing the withheld S−Cu bond. As a consequence, it is evident that sulfide, selenide, and telluride are critically entrapped with Cu, conforming the role of chalcogenide as electropositive enforcers on nearby Cu for facilitating charge-redistribution-induced selective reduction toward formate. Yet, with the continued increase in applied cathodic potential for CuS, the peak intensity at 2470.1 eV decreased while another higher energy peak, corresponding to sulfate, emerged at 2481.7 eV with a relatively stronger white line intensity[59]. This unequivocally demonstrates not only the reduction of Cu as the population of delocalized $Cu^{\delta+}$ hole dropped, but also the oxidation of sulfur to sulfate, characterized by a much lower density of $3p$ electrons. On the other hand, in addition to the constantly reserved Se−Cu and Te−Cu bonds, a minor peak at 1.20 Å, assigned as Se−O and Te−O bonds, was also detected. These indicate the occurrence of slight oxidation on chalcogen atoms. With the application of larger cathodic potentials, the chalcogen atoms commence evacuating the surroundings of Cu, converting into chalcogen oxides such as $SO_4^{2-}$, $SeO_3^{2-}$, and $TeO_2$. Notably, the decline in chalcogenide as a positive charge enforcer could account for the slight decrease in formate selectivity over CO at potentials exceeding −0.7 V. This phenomenon could be attributed to the mitigation of the positive charge on Cu, induced by the charge redistribution effect, resulting in the demotion of Cu−OCHO* population instead. Hence, the substantial cathodic potential promotes the removal of chalcogen atoms and the reduction of Cu, constituting the primary factor contributing to the decline in formate selectivity. Despite the structural change under high cathodic conditions, the rationale of essential Cu−X interaction central to our charge redistribution mechanism remains valid in explaining the loss of formate selectivity and overall catalytic behavior of the CuXs system.

To provide more direct and quantitative evidence, charge density difference, Bader charge analysis, and adsorption energy comparisons for *OCHO and *COOH on CuXs surfaces were performed. Based on the computational results shown in Fig. 5a, b, the adsorption of *OCHO on CuXs exhibits substantially stronger adsorption energies (−1.37 eV, −1.79 eV, and −2.31 eV for CuS, CuSe, and CuTe, respectively) compared to those of *COOH (−0.66 eV, −0.43 eV, and −1.95 eV), indicating that *OCHO is thermodynamically more stable on Cu$^+$ sites in CuXs as induced by the charge redistribution effect. This tendency is consistent with its preferential formation of formate for CuXs. Additionally, charge density difference and Bader charge analyses further reveal that *OCHO adsorption on CuXs induces more pronounced electron transfer from the catalyst surface to the adsorbate (0.41−0.78 |e|) than *COOH (0.02−0.43 |e|). In terms of this enhanced charge transfer, the interaction between adsorbate and substrate is thus strengthened, thereby contributing to the more favorable adsorption energies and greater thermodynamic stability of *OCHO. Collectively, these computational insights quantitatively demonstrate the effective

role of the charge redistribution mechanism in stabilizing key intermediates and directing the formate-selective reaction pathway.

## Discussion

Beyond the typical explanations based on binding energies and transition states for various presumed intermediates and reaction pathways through theoretical calculations, this study provides empirical evidence clarifying the mechanism of copper chalcogenide electrocatalysts in selectively driving $CO_2RR$ toward formate. By integrating various in situ characterizations with $CO_2RR$ performance data, a comprehensive framework for elucidating the entire electroreduction cycle is construed, as depicted in Fig. 5c. Through in situ XAS, Raman, and infrared spectroscopy, we provide direct evidence that chalcogenides stabilize $Cu^{\delta+}$ sites in CuXs, ensuring the absence of $Cu^0$ clusters or $Cu^{\delta+}$−$Cu^0$ mixed-valence species, facilitating the C1 pathway. In situ HERFD-XAS reveals dynamic Cu $4p$ orbital evolution, demonstrating that chalcogen-induced charge redistribution elevates Cu electropositivity, enabling optimal binding interactions with *OCHO and thereby selectively steering the reaction pathway toward formate. CuS, in particular, exhibits competitive performance, achieving a 90% FE with a high formate production rate of 25.4 mmol h$^{-1}$ at −0.6 V. Moreover, its ampere-scale partial current of 1.36 A highlights its potential for large-scale formate production. Contrariwise, CuO counteracts the electropositive nature of Cu, resulting in indiscriminate binding with $CO_2$ molecules and producing both CO and formate. However, as higher potentials are applied, the reduction of $Cu^{\delta+}$ to $Cu^0$ and the subsequent evacuation of chalcogenide ligands become inevitable. This leads to a shift in the product profile, with the emergence of multi-carbon products and over-reduced hydrocarbons such as methane. Ultimately, these findings redefine the fundamental principles governing $CO_2RR$ selectivity and introduce a design strategy for tuning catalyst electronic structure via chalcogen modification. The insights provided herein not only advance the mechanistic understanding of $CO_2RR$ but also lay the groundwork for the rational design of selective and scalable electrocatalysts for sustainable chemical production.

## Methods

### Chemicals and materials
Copper(II) nitrate trihydrate ($Cu(NO_3)_2 \cdot 3H_2O$, Fisher Scientific, analytical reagent grade), sodium borohydride ($NaBH_4$, Fisher Scientific, analytical reagent grade), sulfur powder (S, Showa, 99.5%), tellurium powder (Te, ACROS, 97%), selenourea ($CH_4N_2Se$, Fisher Scientific, 97%), ethylene glycol (EG, Showa, 99.0%), potassium hydroxide (KOH, Fisher Chemical, laboratory reagent grade), and $CO_2$ gas (99.999%) were all used as received without further purification. The ultrapure water (18.2 MΩ cm$^{-1}$) was obtained from a Merck Direct-Q3 ultrapure water purification.

### Catalyst preparation
**Synthesis of CuS.** Copper sulfide (CuS) electrocatalysts were synthesized via a hydrothermal approach. In brief, 4 mmol of $Cu(NO_3)_2 \cdot 3H_2O$ and 8 mmol of sulfur powder were dissolved in 40 mL EG under vigorous stirring for 30 min. Afterward, the solution was transferred into a 100 mL Teflon-lined stainless steel autoclave. The solution was heated at 140 °C for 12 h and cooled to room temperature (25 °C) under ambient temperature. The precipitation was collected by centrifugation (8000 rpm, 10 min), washed with absolute ethanol and ultrapure water multiple times, and dried at 60 °C under vacuum.

**Synthesis of CuSe.** Copper selenide (CuSe) electrocatalysts were synthesized via a hydrothermal approach. In brief, 4 mmol of $Cu(NO_3)_2 \cdot 3H_2O$ and 4 mmol of selenourea were dissolved in 40 mL of water under vigorous stirring for 30 min. Afterward, the solution was transferred into a 100 mL Teflon-lined stainless steel autoclave.

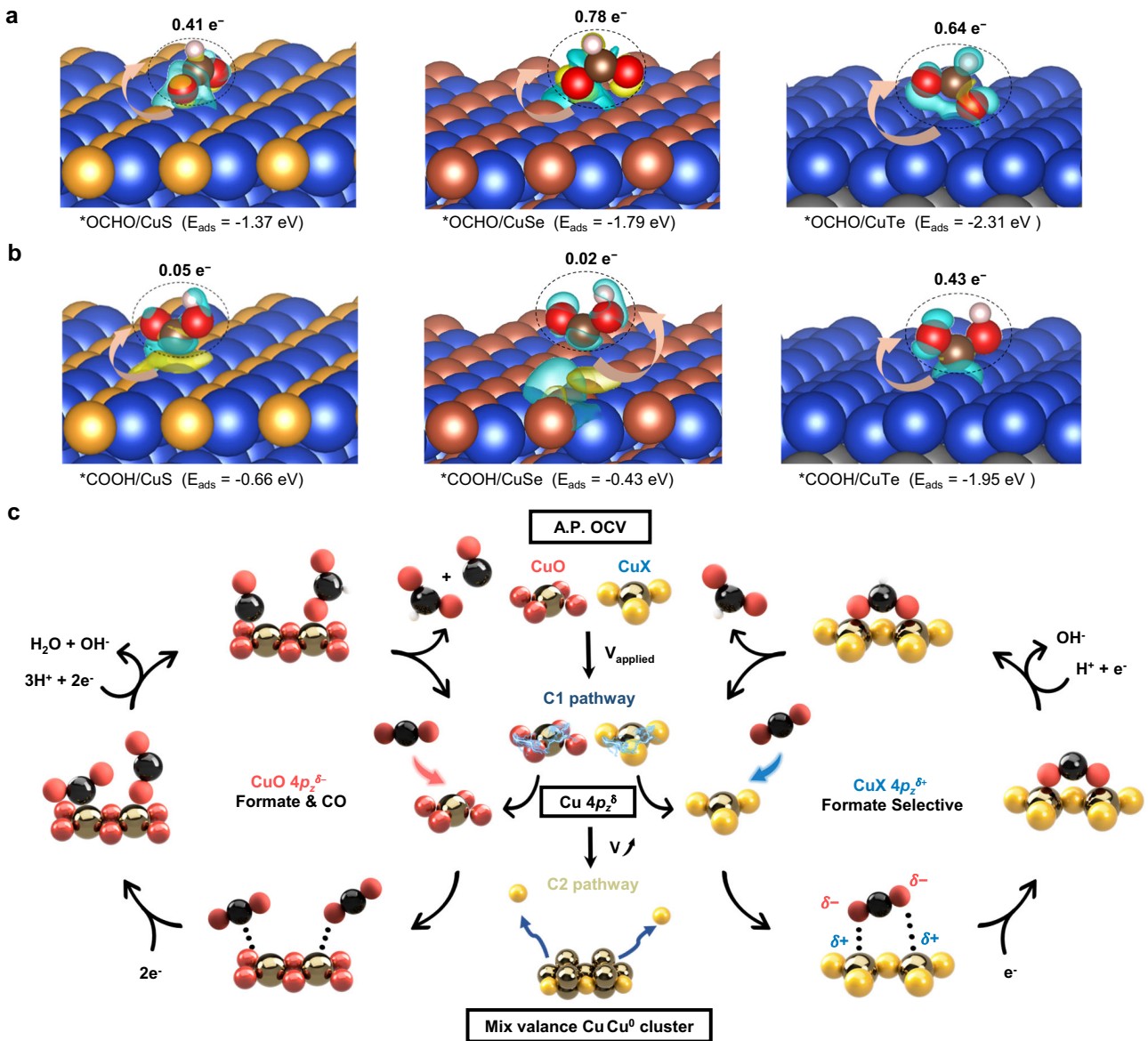

**Fig. 5 | Computational studies and CO₂RR mechanistic understanding.**
**a**, **b** Charge density difference plots for **a** *OCHO/CuXs and **b** *COOH/CuXs configurations. Electron accumulation in cyan and depletion in yellow; the blue, orange, reddish-brown, gray, white, brown, and red spheres represent Cu, S, Se, Te, H, C, and O atoms, respectively. **c** Proposed CO₂RR mechanism over copper-based electrocatalysts for a variety of reduction products and selective reduction to formate.

The solution was heated at 140 °C for 3 h and cooled to room temperature (25 °C) under ambient temperature. The precipitation was collected by centrifugation (8000 rpm, 10 min), washed with absolute ethanol and ultrapure water multiple times, and dried at 60 °C under vacuum.

**Synthesis of CuTe**. Copper telluride (CuTe) electrocatalysts were synthesized via a hydrothermal approach. In brief, 10 mmol of NaBH₄ was dissolved in 20 mL of water under an ice bath. Then, 4 mmol of tellurium powder were added and stirred vigorously for 30 min. Subsequently, a 20 mL aqueous solution of Cu(NO₃)₂·3H₂O (4 mmol) was introduced dropwise to the stirring solution and reacted at 80 °C for 30 min. Afterward, the solution was transferred into a 100 mL Teflon-lined stainless steel autoclave. The solution was heated at 180 °C for 12 h and cooled to room temperature (25 °C) under ambient temperature. The precipitation was collected by centrifugation (8000 rpm, 10 min), washed with absolute ethanol and ultrapure water multiple times, and dried at 60 °C under vacuum.

**Synthesis of CuO**. Copper oxide (CuO) electrocatalysts were synthesized via a hydrothermal approach. In brief, a 40 mL solution of 0.1 M Cu(NO₃)₂·3H₂O (4 mmol) and 0.5 M KOH (20 mmol) were prepared and stirred vigorously for 30 min. Afterward, the solution was transferred into a 100 mL Teflon-lined stainless steel autoclave. The solution was heated at 140 °C for 12 h and cooled to room temperature (25 °C) under ambient temperature. The precipitation was collected by centrifugation (8000 rpm, 10 min), washed with absolute ethanol and ultrapure water multiple times, and dried at 60 °C under vacuum.

### Physical characterization

Powder XRD patterns were collected on a Bruker D2 Phaser diffractometer using Cu K$\alpha$ radiation ($\lambda = 1.54$ Å). Scanning electron microscopy (SEM) images and energy dispersive spectroscopy (EDS) maps were obtained on a field-emission scanning electron microscope (FESEM, JEOL JSM-6700F) equipped with an energy-dispersive X-ray spectroscope (EDX, Oxford Instrument XMAX 150 mm²). The HR-TEM images of the catalysts were collected by using a transmission electron

microscope (TEM, JEOL JEM-2100F) operated at an accelerating voltage of 200 kV with a field-emission gun source. Inductively coupled plasma mass spectroscopy (ICP-MS) result was obtained by an Agilent 7700e (Agilent Technologies). The amount of leaching chalcogen atoms was estimated by the calibration curve of chalcogen standards.

## Electrochemical measurements

The electrochemical measurements were conducted with a Biologic VSP-300 potentiostat using a three-compartment custom-made flow cell with a three-electrode system, a catalyst-sprayed gas diffusion layer (GDL) as the working electrode, Ni foam as the counter electrode, and an Ag/AgCl electrode in 3 M KCl solution as the reference electrode. In all measurements, 10 mg of the prepared catalyst was dispersed in a mixed solution containing 1 mL of isopropanol and 20 $\mu$L of a 5 wt% Nafion/ethanol solution by sonicating for approximately 30 min. The working electrode was prepared by airbrushing the catalyst ink solution on the GDL (100 $\mu$L cm$^{-2}$). For GDL pretreatment in stability tests, a 1 wt% polytetrafluoroethylene (PTFE) emulsion was first sprayed on the GDL to inhibit electrowetting during electrocatalysis, followed by airbrushing with the catalyst ink. A 1 M KOH aqueous solution (pH 13.7) was used as the electrolyte. All electrolytes used in this study were freshly prepared immediately prior to each experiment and used without long-term storage. The electrolyte was circulated in the electrochemical cell at a constant flow rate of 20 mL min$^{-1}$ by using a peristaltic pump; the $CO_2$ flow was at 30 sccm controlled by a mass flow controller at the inlet and a mass flow meter (Bronkhorst) at the outlet to monitor the flow rate preceding and succeeding the electrochemical cell. Two liquid chambers of anode and cathode compartments were separated by a proton exchange membrane (Nafion 117, DuPont) to avoid counter electrode contamination and cathodic product reoxidation. The chronoamperometry (CA) and linear sweep voltammetry (LSV) were carried out with both continuous flow of electrolyte and $CO_2$. All measured potentials in this work were referred to the reversible hydrogen electrode (RHE) and were recorded with $iR$ compensation calculated by Eq. (1).

$$V_{RHE} = V_{Ag/AgCl(3MKCl)} + 0.21V + 0.0592 \cdot 13.7_{(pH \text{ of electrolyte})} - iR \quad (1)$$

Double-layer capacitance ($C_{dl}$) was determined using cyclic voltammetry (CV) measurements in a small potential range of the non-Faradaic region at various scan rates (10–80 mV s$^{-1}$). The $C_{dl}$ value was given by the slope of the plots of scan rate vs anodic/cathodic currents. Electrochemical surface area (ECSA) was then calculated through Eqs. (2 and 3).

$$ECSA = R_f \times S \quad (2)$$

where S is the area of the electrode, and $R_f$ is the roughness factor, which could be estimated by Eq. 2.

$$R_f = \frac{C_{dl}}{C_s} \quad (3)$$

where $C_S$ is the specific capacitance of the sample: we use the general reported values of 40 $\mu$F cm$^{-2}$[60].

## CO$_2$RR product analysis

H$_2$ and other gas components (CO, CH$_4$, and C$_2$H$_4$) were quantitatively analyzed by an online gas chromatography (GC, Agilent 8890, Agilent Technologies) equipped with a thermal conductivity detector (TCD) and flame ionization detector (FID). Argon was used as the carrier gas. The liquid-phase products were quantified using NMR spectroscopy (Bruker Advance III 400 MHz). After a defined reaction time, the liquid samples were withdrawn from the electrochemical cell, mixed with

D$_2$O and dimethyl sulfoxide as the internal standard, and then analyzed by $^1$H NMR spectroscopy using a presaturation technique for water suppression. The total coulomb of electrons (C) for gaseous products of online-GC analysis was derived from Eq. (4).

$$C = i \times \frac{v}{r} \quad (4)$$

where $i$ is the total current during electrolysis, $v$ represents the volume of the GC sampling loop (1 mL), and $r$ is the $CO_2$ flow rate.

The FE of each product was determined by Eq. (5).

$$FE(\%) = \frac{\text{moles of the product} \times n \times F}{C} \times 100\% \quad (5)$$

where $n$ is the number of electrons transferred, $F$ is Faraday's constant (96,485 C mol$^{-1}$), and C represents the total coulomb of electrons passed through the working electrode during the electrolysis.

## In situ X-ray absorption spectroscopy analysis

XAS measurements were recorded in fluorescence yield (FY) mode at room temperature (25 °C) using a Lytle detector. In situ Cu K-edge and Se K-edge X-ray absorption spectra were collected on BL-12B2 at Spring-8 (Japan), Taiwan beamline of the National Synchrotron Radiation Research Center (NSRRC), operated at 8.0 GeV with a current of 100 mA. Ex situ S K-edge and in situ Te K-edge X-ray absorption spectra were collected on beamline TLS-16A and TLS-01C1 of the NSRRC (Taiwan), operated at 1.5 GeV with a current of 360 mA. The above measurements were made in a custom GDE flow cell with Ag/AgCl and Pt wire as reference electrode and counter electrode, respectively. The corresponding raw data was collected from 100 eV before to 600 eV after the edge absorption and processed through edge height normalization and energy calibration by the ATHENA software. The incident energy was calibrated to the first inflection point of the absorption K-edge for a reference foil. EXAFS was obtained using a Fourier transform on $k^2$-weighted oscillations. Phase correction in all EXAFS datasets was not deployed. The curve fitting of corresponding $k^2$-weighted EXAFS spectra was evaluated using Eq. (6) through ARTHEMIS software.

$$\chi(k) = S_0{}^2 \sum_j \frac{N_j}{kR_j^2} S_i(k) F_j(k) e^{-2\sigma_j^2 k^2} e^{-\frac{2R_j}{\lambda_j(k)}} sin\left[2kR_j - \delta_{ij}(k)\right] \left(k = \sqrt{\frac{2m_e}{\hbar^2}\Delta E_0}\right) \quad (6)$$

where $S_0{}^2$ represents the passive electron reduction factor, which accounts for the attenuation of the electron signal due to core-electron screening effects, $R_j$ denotes the distance between the central atom and neighboring atoms in shell $j$, the coordination number $N_j$ is the number of neighboring atoms in shell $j$, $\sigma_j$ is the Debye-Waller factor describing static and thermal disorder, $\Delta E_0$ corresponds to the edge-energy shift, $S_i(k)F_j(k)$ and $\delta_{ij}(k)$ is the ab initio amplitude and phase function for shell $j$, respectively, and $\lambda_j(k)$ is the electron mean free path.

## In situ HERFD-XAS

In situ Kα high-energy-resolution fluorescence detected XAS (HERFD-XAS) experiment in a custom flow cell using a gas diffusion electrode as the working electrode, Ag/AgCl electrode, and platinum wire as reference and counter electrodes, respectively, was performed in 1 M KOH with continuous supply of $CO_2$. All the applied potential was calibrated to a reversible hydrogen electrode (RHE). The incident beam energy was monochromatized by a Si (111) double-crystal monochromator, and a four-bounce channel-cut Si(440) (HRM) was also inserted to enhance the energy resolution to reach 0.3 eV. The scan range was kept in an energy range of 8900–080 eV for Cu K-edge. The fluorescence was split by the analyzer crystal so that the signal

would be collected by a silicon drift detector (XR-100CR Si-PIN X-ray detector) in partial-fluorescence-yield mode on BL-12XU at SPring-8 (Japan).

### In situ X-ray scattering technique analysis

The in situ X-ray scattering/diffraction (XRD) measurement with a wavelength of 0.6894 Å (18 keV) was carried out under a customized liquid cell with Ag/AgCl and Pt wire as reference electrode and counter electrode, respectively. For ex situ XRD, identical instrumentation was used without the liquid cell. All above measurements were performed at the BL-12B2 of Spring-8 (Japan), Taiwan beamline of the National Synchrotron Radiation Research Center (NSRRC), and TLS-01C2 beamline of NSRRC (National Synchrotron Radiation Research Center), Taiwan.

### In situ Raman characterization

UniNano UNIDRON was adopted to measure in situ Raman spectra. A $50\times$ objective lens was used to focus the He-Ne laser with 633 nm on the sample, where the laser spot size is $1\,\mu m^2$. The measurement was performed under an exposure time of 5 s with an accumulation number of 30 times by illuminating a 2.5 mW laser power. The in situ experiments were performed in a customized flow cell designed specifically for Raman measurements with Ag/AgCl and Pt wire as reference electrode and counter electrode, respectively.

### In situ SEIRAS characterization

In situ SEIRAS was recorded in the Kretschmann ATR configuration using a Vertex 70 FTIR (Bruker) spectrometer equipped with an HgCdTe (MCT) detector[61] and a single-reflection accessory VeeMAX III (PIKE). The incident angle was set to 62.5°. The spectrometer purged $N_2(g)$ through the measurement. The internal reflection element (IRE) was a 60° face-angled Si crystal (PIKE). The measurements were performed in Jackfish SEC J1 Cell (Jackfish SEC) with Ag/AgCl and graphite rod as reference electrode and counter electrode, respectively. The graphite rod was used to avoid any possible contamination from the Pt electrode[62]. Before the measurement, Au film with a thickness of about 20 nm was coated on the principal reflecting plane of the IRE by physical vapor deposition (PVD) with a deposition rate of $0.4\,\text{Å s}^{-1}$ using $Ar^+$ as bombardment source. The deposition rate and thickness were monitored using a quartz crystal microbalance. The catalyst ink of $100\,\mu L$ prepared in the previous section was dropped onto the Au-coated reflecting plane as the working electrode. IR reference spectra were obtained at OCV in a $CO_2$-saturated 1.0 M KOH electrolyte. The spectra were sequentially acquired with a spectral resolution of $8\,\text{cm}^{-1}$ at around 12 s for chronoamperometry measurements. The spectrometer was operated in kinetic mode (40 kHz). A single-beam spectrum collected at the starting potential was used as the reference spectrum. All ATR-SEIRAS spectra were reported in absorbance units defined as $A = -\log(I/I_O)$, where $I$ and $I_O$ denote the light intensity for the sample and reference single-beam spectra, respectively.

### DFT computational details

All spin-polarized electronic structure calculations were conducted using the plane-wave-based Vienna ab initio Simulation Package (VASP)[63,64]. The calculations of electron–ion interactions and exchange–correlation effects are conducted using the generalized gradient approximation (GGA), specifically with the Perdew–Burke–Ernzerhof (PBE) functional employed as its representative formulation[65,66]. The electronic wave functions were expanded using a plane-wave basis set with a kinetic energy cutoff of 400 eV, while the interaction between valence and core electrons was treated using the projector augmented-wave (PAW) potentials[67]. For the purpose of density of states (DOS) calculations, all bulk crystal structures obtained from single-crystal XRD measurements conducted in our

laboratory were employed in single-point energy calculations without further geometry optimization, and Brillouin zone integration was subsequently performed using a Monkhorst-Pack $k$-point mesh with a resolution of $12\times12\times12$ to ensure accurate electronic structure sampling[68]. For slab surface models (CuX systems, where X = S, Se, and Te), charge density difference, Bader charge, and adsorption energy calculations were carried out using slab geometries with a vacuum layer of approximately 17 Å to eliminate interactions between periodic images, and the Brillouin zone was sampled using a Monkhorst-Pack k-point mesh of $4\times4\times1$. In particular, we focused on the detailed orbital-resolved PDOS of the Cu $3d$ orbitals (including $d_{z2}$, $d_{x2-y2}$, $d_{xy}$, $d_{yz}$, and $d_{xz}$), as well as the outermost $p$ orbitals (including $p_x$, $p_y$, and $p_z$) of the O, S, Se, and Te atoms. All PDOS values were normalized to the number of atoms in the periodic system, and plotted as a function of energy relative to the Fermi level, expressed as $DOS(E) = f(E - E_f)$, where $E - E_f$ represents the energy offset from the Fermi energy ($E_f$). All atomic positions of DFT models for copper chalcogenides are provided in Supplementary Data 1.

## Data availability

The datasets that support the findings of this study are presented in the text and Supplementary Information. The atomic positions of all DFT models are provided in the Supplementary Data 1. Source data are provided with this paper. The source data are available via Zenodo at https://zenodo.org/uploads/17082229. Source data are provided with this paper.

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

## Acknowledgements

We acknowledge support from the National Science and Technology Council, Taiwan (contracts no. NSTC 113–2123-M-002 -005, NSTC 114-2124-M-002-014, 114-2113-M-034-003, and 112-2926-I-002–513-G), the MCUT Formosa Center, and from the National Taiwan University (NTU-CC-113L893304). We thank the mass spectrometry technical research services from Consortia of Key Technologies, National Taiwan University; Ms. C.-Y. Chien of the National Science and Technology Council (National Taiwan University) for assistance with FE-TEM experiments; and the interdisciplinary project of NSRRC for providing assistance with the synchrotron-based X-ray experiments.

## Author contributions

H.M.C. and F.-Z.T. conceived the project. F.-Z.T. and P.-J.L. performed the catalyst synthesis and the majority of characterization. F.-Z.T. performed the electrochemical measurements, in situ synchrotron spectroscopy experiments, data analysis, and wrote the paper. W.-J.C. performed the in situ Raman spectroscopy experiment. Y.-A.L. and P.-J.L. performed the in situ SEIRAS experiment. C.-S.H. assisted in the synchrotron spectral data analysis and simulation. Y.-H.C. contributed to the electrochemical data analysis. S.-C.L. and Y.-C.C. contributed to the spectral analysis. C.-S.H. and S.-W.H. supported the setting up and interpretation of the synchrotron spectroscopy experiments. H.-L.C. conducted the DFT calculations and contributed to theoretical insights. H.M.C. supervised the overall project and co-wrote the paper. All authors discussed the results and contributed to the final paper.

## Competing interests

The authors declare no competing interests.
