## [Transparent Peer Review file · Nature Communications]

Charge redistribution dynamics in chalcogenide-stabilized cuprous electrocatalysts unleash ampere-scale partial current toward formate production

Corresponding Author: Professor Hao Ming Chen

Version 0:

Reviewer comments:

Reviewer #1

(Remarks to the Author)

This manuscript by Hao Ming Chen et al. studies the CO₂ reduction reaction on a series of copper chalcogenides and proposes that cuprous can be stabilized, which enables the formate formation at a relatively higher overpotential compared with CuO counterpart. Albeit this work does not show significant advancement in terms of performance, the in-situ studies might provide useful insights into CO₂-to-formate electrosynthesis. At this point, I cannot support the publication on Nature Communication because this work lacks materials evolution analysis along the reaction time and lacks post-electrolysis materials characterization. For more details:

- Copper chalcogenides degradation under reductive potential is a known phenomenon (e.g., ACS Catalysis, 2017, 837), resulting in copper metal. In this paper, the authors used in-situ XANES etc., as evidence to show that Cu(I) dominates in CuSe and CuTe.
- These results may not reflect the longer-term Cu state in these chalcogenides. One possibility is that Se or Te can slow down the Cu metal formation but do not stabilize Cu(I).
- This is not just affecting the applied perspectives of CuSe and CuTe. The outcome of the in-situ studies may also be changed by the time scale of the measurements.

My recommendations:

- Should the authors provide a post-electrolysis (at least 2 hours) materials characterization for CuO, CuS, CuSe and CuTe.
- The authors should show evidence that Cu(I) is stabilized when conducting the in-situ measurement, i.e., the Cu(II):Cu(I):Cu(0) ratio under a certain potential is not changed over time, in a period of 30 minutes.
- The authors may supplement a Pourbaix diagram of CuS, CuSe and CuTe.

Reviewer #2

(Remarks to the Author)

This paper investigates chalcogenide-stabilized cuprous electrocatalysts for the electrocatalytic reduction of carbon dioxide (CO₂RR) to formate. It proposes that high-selectivity formate generation can be achieved through a charge redistribution mechanism induced by chalcogen anions, supported by various in situ characterization techniques. While the study offers certain novel insights, the rationale for employing expensive in situ measurements appears insufficient. Specifically, the authors do not clearly articulate the concept of charge redistribution or evaluate its impact on formate selectivity across different CuX catalysts. Additionally, certain experimental details, data analysis, and discussion depth could be improved. Overall, this work is potentially publishable pending revision.

Major Points:

The performance tests were conducted in a flow cell, while XAS measurements were performed in a batch H-cell. Given that the H-cell typically operates at lower currents, it may not accurately reflect the catalyst's oxidation state in the flow cell. The authors should address the differences in current and their potential impact on the results, as this could challenge the

proposed mechanism and conclusions.

The charge redistribution mechanism is proposed to link chalcogen anion doping in Cu with formate selectivity. However, the authors do not adequately explain why CuS exhibits superior performance within a narrow potential window, while CuSe and CuTe favor hydrogen evolution over formate production. A more detailed mechanistic explanation is needed.

The stability test was conducted at ~60 mA/cm² for only 8 hours. Given that CO₂RR to formate has been reported to achieve stability for over 5000 hours (Nature 626, 86–91 (2024)), the authors should justify the short duration. Additionally, constructing Pourbaix diagrams for different CuX catalysts could help correlate their stability under CO₂RR conditions with the proposed mechanism.

Minor Points:

The main text exceeds 6000 words. The authors should consider condensing the reasoning and discussions to emphasize key findings, particularly those not previously reported.

The discussion of as-prepared catalysts (Fig. 1a-1e) is overly detailed and less relevant. These sections should be moved to the SI, allowing the authors to focus on the derived catalysts and their structural roles in CO₂RR selectivity.

The paper attributes the peak at 2800–3000 cm⁻¹ to HOO- adsorption. However, a strong signal is also observed on CuO, which primarily produces C₂+ products, with a formate FE of only 27%. The authors should justify the assignment of this species.

The CO peak around 2000 cm⁻¹ in the Raman spectrum for CuS should be included for a more comprehensive comparison.

Critical details and justifications of in situ testing under CO₂RR are missing. For example, in situ SEIRAS was performed by purging CO₂ into 1 M KOH, which ultimately forms (bi)carbonate. The authors should explain why (bi)carbonate electrolyte was not used directly.

The authors might consider incorporating DFT calculations to validate the proposed charge redistribution mechanism for formate production on Cu-based catalysts.

There are several typos that need correction. For instance, on page 5, 'Cu-S, Cu-S, and Cu-Te' should be 'Cu-S, Cu-Se, and Cu-Te'; on page 6, 'As depicted in Figure 2c for CuS' should be 'As depicted in Figure 2c for CuO'. These and other errors should be addressed.

Reviewer #3

(Remarks to the Author)

In this work, the authors used a series of copper chalcogenides and CuO to study the underlying mechanism of highly selective generation of formic acid. The authors detected the evolution of intermediates and changes in catalyst structure during the reaction by a set of in situ characterization techniques. The analysis of the results revealed the influence of chalcogenides on the structure, chemical state and electron distribution, which led to the improvement of formic acid selectivity. This paper is interesting and can be published after addressing the following points properly.

1. On Page 6, line 166, the CuS catalyst only can maintain the FE of formate above 85% for 8 hours. Can the authors explain why the stability of the catalyst can only be maintained for such a short time? Is it because of salt precipitation at the gas diffusion layer or a drastic change in the structure of the catalyst? If it is caused by the structural change, please provide structural data after stability testing and time-resolved in situ XAS data to prove it.
2. On Page 6, line 179, the author claims that the doublet peak at 1440 and 1460 cm⁻¹ is verified as the identity of oxo-bridged site of *OCHO. However, according to the references cited by the author (Angew. Chem. 2019, 131 (5), 1359–1363), this peak is attributed to the vibration peak of *OCHO on Ag. If the author wants to show that this peak also represents the adsorption peak of *HCHO on Cu, they should provide isotope labeling experiments to prove that this peak belongs to the -O-(CH)-O vibration on Cu.
3. On Page 6, lines 189-192, according to the description in the manuscript and supplementary information, the SEIRAS were collected in 1 M KOH saturated with CO₂. However, it is usually impossible to obtain SEIRAS signals under strong alkaline condition (unless the special designed SEIRAS cell is used). This is because KOH will react with CO₂ in the electrolyte and eventually converted into KHCO₃, causing the electrolyte composition to change. Therefore, the signals actually collected is under KHCO₃ conditions rather than KOH. The authors' conclusions regarding SEIRAS are therefore not credible.
4. On Page 8, lines 245-247, the authors claim that whether it is CuS, CuSe, CuTe or CuO, these catalysts maintain their original phases and do not undergo obvious structural transformation or formation under electrochemical reduction conditions. Obviously, this result is incomprehensible and unbelievable. First, previous studies (e.g., ChemSusChem 2023, 16, e202300879) have shown that under reducing conditions, CuO and CuS will undergo O and S leaching from the lattice, undergo drastic structural changes, and eventually transform into other phases. Secondly, on page 9, line 286, the author also believes the evacuation of sulfide in CuS cause the formation of Cu₀ cluster. Based on the mentioned above results, it is impossible for CuS and CuO not to undergo phase change at negative potential. However, the authors only explained that the unusual in situ XRD results were caused by the structural variations not in long-range order but in short-range order.

Thus, this explanation is not convincing, and the author should provide a more reasonable and comprehensive explanation. 5. On Page 7, line 194, the author explains the formate-related peak could not be distinctly observed due to peak overlap of H–O–H bending mode of adsorbed water at around 1300 ~ 1600 cm^{-1} . However, the author did not explain the disappearance of the characteristic peak of adsorbing formate around 2800 ~ 3000 cm^{-1} in Figure 2d-e. Furthermore, the disappearance of the peak in this range is unlikely to be caused by the peak overlap of H-O-H stretching mode of adsorbed water because it usually appears above 3000 cm^{-1} .

6. Page 9, lines 266-267, the authors believe that Cu^{1+} as the dominant specie directing selectivity toward C1 products. Nevertheless, at low potential (-0.4 V ~ -0.5 V), Cu^{2+} accounts for a large proportion or even the dominant specie in CuS and CuSe. However, under this condition, C1 is still the main product. The author should explain it.

7. As shown in Figure 2a and 2c, the broad band at 2800 ~ 3000 cm^{-1} in Figure is not HOO–ads but HCOO–ads, the author should change it.

8. On Page 6, line 193, “As depicted in Figure 2c for CuS”. The author should change Figure 2c to Figure 2d.

9. In Figure 4b, the “-06V” should change to “-0.6V”.

10. Page 11, line 352, the author claims that “the more pronounced splitting of the 4p orbitals in CuO reflects a stronger interaction between Cu and O ligands”. This sentence is confusing. If the Cu-O interaction is stronger, the Cu-O bond is more stable than other Cu-Xs and will not easily destroy at negative potential. But in situ XAS results show that O is more easily detached from the lattice at negative potential than other chalcogens, which indicates that the Cu-O interaction should be weaker. Please give the more depth explanation.

11. Page 13, lines 402-403, the author conducts further investigations into chemical environments through ex situ S K-edge, in situ Se K-edge, and in situ Te K-edge XAS. The author also needs to provide in situ O K-edge to study the chemical environment of CuO.

12. In Figure 4b, under -0.6V conditions, the spectra of CuXs and CuO changed to varying degrees compared to their initial states. It can be seen from the Figure that the degree of change is in the order of $\text{CuO} > \text{CuS} > \text{CuSe} > \text{CuTe}$. Could this difference in the extent of change be due to different anions having different levels of loss? Because under electrochemical reduction conditions, O and S are more easily lost from the lattice than Se and Te, causing great structural changes. The author may need to provide R-space of Figure 4b to supplement the explanation of whether this change is caused by a drastic change in the coordination structure.

Reviewer #4

(Remarks to the Author)

The authors present a compelling study on CO_2 reduction (CO_2RR) using Cu-based catalysts. By employing a comprehensive set of in situ and operando techniques—including X-ray absorption spectroscopy (XAS), high-energy-resolution fluorescence-detected XAS (HERFD-XAS), operando Raman, and infrared spectroscopy—they provide direct evidence that Cu–chalcogen interactions stabilize Cu^+ species, preventing their over-reduction to metallic Cu^0 . This stabilization also modulates CO_2 adsorption and the binding of reaction intermediates. Among the materials studied, CuS shows the highest selectivity, proving well with its potential for industrial-scale application.

The characterization work is thorough, and the conclusions are supported by the data. The paper clearly demonstrates the charge redistribution mechanism in copper chalcogenides. The discussion is well-structured and convincing, making the manuscript suitable for publication in Nature Communications.

Two minor suggestions:

Please improve the quality of Figure 1a, as the XRD data are currently difficult to discern.

For the HERFD data, a projected density of states (p-DOS) calculation could enhance the discussion around the Cu 4p orbitals. If possible, I recommend including this analysis in the Supporting Information.

Reviewer #5

(Remarks to the Author)

H. M. Chen*, et al reported the effect of charge redistribution in the Cu compounds-based CO_2 reduction reaction (CO_2RR) catalysts on steering the CO_2RR selectivity between the formate and C2 products. Authors compared the CO_2RR of Cu oxide (CuO) and Cu chalcogenide-based compounds (CuX) such as CuS, CuSe, CuTe. Comprehensive studies with real time characterizations such as in-situ Raman, SEIRAS for CO_2RR intermediate detection, in-situ XAS and HERFD-XAS for oxidation states, coordinates structures analysis successfully supports the importance of charge redistribution on selectivity control. However, authors need to consider the effect of reconstruction on not only for charge distribution but also the surface structure change, which can affect the bindings of CO_2RR intermediates. Therefore, I recommend the major revision of this manuscript for Nature Communications. Details are as follows:

1. Authors explained that partially positive surface oxidation states of Cu during CO_2RR in Cu chalcogenides catalysts can induce formate production. How can the oxidation states of Cu remain positive during CO_2RR under reductive potential? Unlike the reconstruction of Cu oxide, how does the reconstruction of Cu chalcogenides proceed? Authors need to verify the content of S, Se, Te in the catalysts as well as the dissolved anions during CO_2RR according to the applied potential and reaction time.

2. In the Figure S1, it seems that the size, morphology, and surface structures of CuS, CuSe, CuTe, and CuO are different.

These parameters are related with surface area (Figure S9) and this can also affect the degree of reconstruction. I recommend investigating the reconstruction and CO₂RR studies with catalysts with similar structures.

3. Authors need to investigate how the surface structure of catalysts changes during CO₂RR such as surface facet, crystallinity, surface roughness which can be altered by reconstruction. The current manuscript mainly focuses on the oxidation states; however, those structure factors should be considered for a more comprehensive understanding.

Version 1:

Reviewer comments:

Reviewer #1

(Remarks to the Author)

The authors have added additional experimental results and discussions to address my concerns. I now support the publication as is.

Reviewer #2

(Remarks to the Author)

The authors' response has addressed most of my concerns regarding this work. However, I still have two remaining questions that I would like the authors to clarify.

1. The concept of "charge redistribution" as the origin of formate selectivity is central to this work, but its definition and physical basis remain somewhat vague. The current support from HERFD-XAS and PDOS analysis is informative, yet largely qualitative. I suggest the authors strengthen this claim with more direct evidence, such as charge density differences, Bader charge analysis, or adsorption energy comparisons between *OCHO and *COOH. The distinction between Cu⁺ stabilization and intermediate pathway preference should also be made clearer.

2. While the added long-term XAS, XRD, and HR-TEM data demonstrate good structural retention for CuX catalysts, the observed sulfur leaching in CuS (~20%) raises questions about possible local coordination changes. Given the mechanistic emphasis on Cu–X interactions, even subtle reconstruction could influence selectivity. The authors should better contextualize how such evolution—particularly under higher cathodic bias—affects the validity and scope of the proposed mechanism.

Reviewer #3

(Remarks to the Author)

The authors have addressed all my concerns and the manuscript has been improved. I thus recommend its publication as it is.

Reviewer #4

(Remarks to the Author)

This study investigates the charge redistribution mechanism in copper chalcogenides of several compositions, used as electrocatalysts for the selective CO₂ reduction reaction (CO₂RR) toward formate production. The authors employ several valuable characterization techniques, such as EXAFS, HERFD, XRD, and HRTEM. To support the experimental findings, the authors also perform DFT calculations and analyze the projected density of states (p-DOS) to help interpret the XANES spectra.

The paper has been revised and the authors have addressed all reviewer comments.

The results provide direct evidence that chalcogenides stabilize cuprous species, preventing their over-reduction to metallic Cu, which is detrimental to the reaction as it favors the formation of multi-carbon products. Moreover, in situ HERFD-XAS reveals the dynamic evolution of Cu 4p orbitals, which the authors interpret using p-DOS analysis.

The conclusions are supported by the data and analysis provided by the authors.

I consider the paper suitable for publication as it is.

Reviewer #5

(Remarks to the Author)

The revised manuscript now addresses the concerns from this reviewer. I recommend the acceptance of this manuscript in Nature Communications.

Version 2:

Reviewer comments:

Reviewer #2

(Remarks to the Author)

The authors have addressed my concerns. The manuscript is now well-prepared and suitable for publication in its current form.
